# Calcium-Silicate-Incorporated Gellan-Chitosan Induced Osteogenic Differentiation in Mesenchymal Stromal Cells

**DOI:** 10.3390/polym13193211

**Published:** 2021-09-22

**Authors:** Krishnamurithy Genasan, Mohammad Mehrali, Tarini Veerappan, Sepehr Talebian, Murali Malliga Raman, Simmrat Singh, Sasikumar Swamiappan, Mehdi Mehrali, Tunku Kamarul, Hanumantha Rao Balaji Raghavendran

**Affiliations:** 1National Orthopaedic Centre of Excellence in Research and Learning (NOCERAL), Tissue Engineering Group (TEG), Department of Orthopaedic Surgery, Faculty of Medicine, University of Malaya, Kuala Lumpur 50603, Malaysia; krishna_82@um.edu.my (K.G.); tariniv90@gmail.com (T.V.); mrmurali08@gmail.com (M.M.R.); simmratsingh@gmail.com (S.S.); 2Faculty of Engineering Technology, Department of Thermal and Fluid Engineering (TFE), University of Twente, 7500 AE Enschede, The Netherlands; mehrali.gary@gmail.com; 3Faculty of Engineering, School of Chemical and Biomolecular Engineering, The University of Sydney, Sydney, NSW 2006, Australia; sepehr.tlb@gmail.com; 4Nano Institute (Sydney Nano), The University of Sydney, Sydney, NSW 2006, Australia; 5Materials Chemistry Division, School of Advanced Sciences, VIT University, Vellore 632014, Tamil Nadu, India; ssasikumar@vit.ac.in; 6Department of Mechanical Engineering, Technical University of Denmark, 2800 Kgs Lyngby, Denmark; meme@mek.dtu.dk; 7Advanced Medical and Dental Institute (AMDI), University Sains Malaysia, Bertam, Kepala Batas 13200, Penang, Malaysia; 8Faculty of Clinical Research, Central Research Facility, Sri Ramachandra Institute of Higher Education and Research Porur, Chennai 600116, Tamil Nadu, India

**Keywords:** gellan, chitosan, bone, calcium silicate, osteoinduction, bone tissue engineering

## Abstract

Gellan-chitosan (GC) incorporated with CS: 0% (GC-0 CS), 10% (GC-10 CS), 20% (GC-20 CS) or 40% (GC-40 CS) *w*/*w* was prepared using freeze-drying method to investigate its physicochemical, biocompatible, and osteoinductive properties in human bone-marrow mesenchymal stromal cells (hBMSCs). The composition of different groups was reflected in physicochemical analyses performed using BET, FTIR, and XRD. The SEM micrographs revealed excellent hBMSCs attachment in GC-40 CS. The Alamar Blue assay indicated an increased proliferation and viability of seeded hBMSCs in all groups on day 21 as compared with day 0. The hBMSCs seeded in GC-40 CS indicated osteogenic differentiation based on an amplified alkaline-phosphatase release on day 7 and 14 as compared with day 0. These cells supported bone mineralization on GC-40 CS based on Alizarin-Red assay on day 21 as compared with day 7 and increased their osteogenic gene expression (RUNX2, ALP, BGLAP, BMP, and Osteonectin) on day 21. The GC-40 CS–seeded hBMSCs initiated their osteogenic differentiation on day 7 as compared with counterparts based on an increased expression of type-1 collagen and BMP2 in immunocytochemistry analysis. In conclusion, the incorporation of 40% (*w*/*w*) calcium silicate in gellan-chitosan showed osteoinduction potential in hBMSCs, making it a potential biomaterial to treat critical bone defects.

## 1. Introduction

As lifespan increases, injuries related to bones are common among elderly individuals, compromising their activities of daily living (ADLs) and posing lifestyle and economic challenges [1]. Despite the fact that advancements have been achieved in bone tissue engineering to date, there are drawbacks to the current treatments [2]. This has led to the development of the biotechnological sector, with the aim of establishing biomaterials for bone tissue engineering. Furthermore, the demand for guided bone regeneration procedures involving bone grafts and membranes are expected to soar from 4.4% to 9.5% from 2018 to 2026 [3]. The international revenue based on treatment for bone defect repairs is predicted to double from 2018 to 2025 [4].

In line with this, a considerable amount of attention has been paid to the use of biodegradable polymers for bone tissue engineering. Furthermore, these polymers were proven to induce the osteogenic differentiation in bone marrow derived cells [5]. Different types of decomposable polymers have been used in the fabrication of porous scaffolds. Particularly, much focus has been given to polymers derived from natural sources owing to its chemical versatility and extracellular matrix that support excellent cellular interactions [6].

Chitosan is a well-examined polysaccharide that has a β-link between glucosamine and N-acetyl glucosamine. It has biological properties such as antibacterial activity, biodegradability, and tissue compatibility [7,8]. Research has indicated that chitosan scaffolds facilitate bone regeneration in in vitro and animal models [9]. Gellan gum is an easily degradable polysaccharide, isolated from *Pseudomonas elodea*; it is non-toxic, and biologically compatible. It has recently attracted attention because it has low pH and superior gelation properties [10]. In general, gellan gum (polysaccharide with anionic properties) can interact with chitosan (a cationic polysaccharide) ionically to make a gellan gum–chitosan complex, a feasible material composition for tissue engineering [11,12]. Previous data have shown chitosan and gellan enhance intestine tissue mucoadhesion and improve the release of bioactive components in a sustained pattern [13,14].

For years, bio-materials such as cement-based ceramics have been used in the fabrication of porous bio-scaffolds because they provide superior porosity and suitable morphology for bone tissue engineering [15]. Previous data have showed that calcium silicate (CS) is capable of inducing osseointegration (in vivo), and its action is credited with hydroxyapatite nucleation, triggered by the dissolution of silicate and calcium ions. It has osteoinductive properties that regulate specific cellular responses through downstream signaling as evidenced in some of the studies [16,17,18]. Reports have shown that CS can prevent the expression of inflammatory markers in human dental cells and enhance the bone mineral secretion in many primary cell types [19]. However, we anticipate that a combination of calcium silicate and gellan-chitosan-based biomaterials, may enhance osteogenic differentiation and biocompatibility properties in human mesenchymal stromal cells.

Bone maintenance and remodeling were implemented using different cells, such as bone-lining and mesenchymal stromal cells [20]. Mesenchymal stromal cells are found in the population of bone marrow within bone cavity and periosteum tissue. Earlier studies reported the role of 3D coragraf and TCP-chitosan scaffolds with fucoidan in supporting the differentiation potential of bone marrow stromal cells. Herein, we explore the effect of 3D gellan-chitosan scaffolds incorporated with 0%, 10%, 20%, and 40% *w*/*w* of calcium silicate, annotated as GC-0 CS, GC-10 CS, GC-20 CS, and GC-40 CS, in supporting hBMSC attachment, viability/proliferation, and osteogenic differentiation potential. 

## 2. Materials and Methods

### 2.1. Synthesis of Calcium Silicate

The method for synthesis of calcium silicate was adopted from our previous research [21]. Briefly, 18–20 mL of calcium nitrate tetrahydrate (Ca (NO_3_)_2_·4H_2_O) 0.2 M in distilled water was prepared by stirring for 25 min (pH = 12, with NaOH), followed by the addition of 20 mL of 0.2 M sodium metasilicate nonahydrate (Na_2_SiO_3_·9H_2_O) solution (Cat:13517-24-3, Sigma-Aldrich, Darmstadt, Germany) in a dropwise manner into the first solution. The resultant suspension was mechanically stirred at room temperature (1 h) to obtain a homogeneous suspension. The suspension was transferred into a Teflon-lined stainless steel (60 mL) autoclave, heated to 200 °C for 24 h, and then naturally cooled to room temperature. The filtered suspension was washed away several times by centrifugation and resuspension with double-distilled water, which was done after hydrothermal treatment, and powders were dried for 24 h at 100 °C.

### 2.2. Biomaterial Fabrication

The biomaterial fabrication was established in our laboratory [21]. To prepare the scaffolds, different concentrations of calcium silicate (0%, 10%, 20%, 40% *w*/*w*) (Cat: 1344-95-2, Sigma-Aldrich, Darmstadt, Germany), synthesized as previously described [21], were dispersed in a 1% acetic acid solution, to which 3% (*w*/*w* %) of chitosan (MW:160 kDa) (Cat: 9012-76-4, Sigma-Aldrich, Darmstadt, Germany) was added. The chitosan/calcium silicate with the gellan gum solution at a 50/50 ratio (*v*/*v*) was allowed to mix to form a precursor, which was viscose in nature. The slurry was poured into cell plates (24-well), followed by freeze-drying (Model: 7740020, Labconco, Kansas City, MO, USA), to construct the materials. 

### 2.3. Physicochemical Characterization

The physicochemical characterization was performed as per standard procedures in our laboratory [22]. A D8 Advance X-Ray diffractometer (Bruker-AXS, Billerica, MA, USA) was used to record Bragg peak XRD patterns for GC-0 CS, GC-10 CS, GC-20 CS, and GC-40 CS. The Ni-filtered monochromatized (CuKα radiation (λ = 1.54056 Ǻ) at 40 kV and 40 mA at 25 °C) was used to operate the diffractometer, with a scanning rate of 0.1 degree s^−1^. Peak diffraction (Bragg) patterns were plotted in the range of 5 to 100 2θ°. The type of chemical functional group present in the composite scaffolds was examined using FTIR, Spectrum 400 (PerkinElmer, Waltham, MA, USA), employing a KBr pellet disc technique. The KBr powders and pellet were mixed (1:10 ratio), and the mixture was pressed at 10 bar pressure to achieve a tablet form. The spectra were recorded in the range of (400–4000 cm^−1^), with a scan rate of 4 cm^−1^ over 130 scans. The Brunauer-Emmett-Teller method (BET ASAP2020, TRISTAR II, Norcross, GA, USA) was executed to analyze the specific surface area of the materials from the nitrogen adsorption-desorption isotherms.

### 2.4. hBMSCs Isolation and Seeding

Approval was granted by the University of Malaya Medical Centre Medical Research Ethics Committee (UMMC-MREC) to conduct this study (Ethics No: 967.10). Before obtaining human bone marrow aspirates, written consent was obtained from 50- to 70-year-old arthroplasty subjects. The hBMSC isolation and characterization, to confirm the plastic-adherent, phenotypic surface marker expression (lineage positive: CD44, CD73, CD90 and CD105 and lineage negative: CD45 and CD34) and the lineage differentiation potential (Osteogenic, adipogenic, and chondrogenic), were performed in our laboratory [22,23]. Briefly, passage-1 cells were detached using trypsin from culture flasks. The isolated cells were seeded in gamma-irradiated (aseptic) GC-0 CS, GC-10 CS, GC-20 CS, and GC-40 CS scaffolds at a cell density of 1 × 10^3^ cells/mL in a 96-well plate. Baseline control was established without cells. The cell scaffold constructs were supplemented with DMEM-LG (Cat: 11885084, Thermo Fisher Scientific, Waltham, MA, USA) culture media, containing 1% penicillin and streptomycin (Cat: 15140148, Thermo Fisher Scientific, Waltham, MA, USA) and 10% fetal calf serum (Cat: A4766801, Thermo Fisher Scientific, Waltham, MA, USA) and incubated at 37 °C in 5% CO_2_ with a humidity of 95%. The culture medium was changed at 24 h intervals.

### 2.5. Scanning Electron Microscopy Analysis

Scanning electron microscopy (SEM) (FEI Quanta 200F, Hillsboro, OR, USA) topography analysis was executed to observe the distribution of hBMSCs on all composite scaffolds using our established method [24]. Briefly, scaffold samples were postfixed using 2.5% formalin and subjected to a dehydration procedure using a graded series of ethanol-water ratios (40–100%) and maintained in a fume hood for drying at room temperature. Using sputter coating machine, scaffolds were coated with platinum and scanned using scanning electron microscopy (Phenom G2).

### 2.6. Cell Proliferation and Viability Assay

The cell proliferation and viability assay was executed to the cell-seeded GC-0 CS, GC-10 CS, GC-20 CS, and GC-40 CS scaffolds based on our established method on days 0, 7, 14, and 21 [22]. Briefly, colorimetric indicator Alamar Blue (AB) cell proliferation and viability assay (Cat: DAL1100, Invitrogen, Waltham, MA, USA) was directly added into the media in all preparations at a final concentration of 10% and incubated for 10 h. After incubation, 100 μL of medium from each sample was transferred into a 96-well plate in duplicate. Absorbance in each well was measured at 570 and 600 nm (reference wavelength) using a microplate reader (BioTek Epoch, Winooski, VT, USA). The corrected absorbance readings were calculated by subtracting the individual reference wavelength from the respective measured wavelength and presented as mean ± standard deviation (SD).

### 2.7. Alkaline Phosphatase (ALP) Activity

The alkaline phosphatase (ALP) release of cell-seeded GC-0 CS, GC-10 CS, GC-20 CS, and GC-40 CS fabricated materials was measured at days 0, 7, and 14 from the culture media using the manufacturer’s protocol (Cat: ab83369, Abcam, Waltham, MA, USA) for an ALP colorimetric assay kit. A total of 50 μL of media from cell-seeded scaffolds (*n* = 3), in duplicate, was mixed with 30 μL of p-NPP substrate. The aspirates were then added with assay buffer, to make a final volume of 130 μL, and they were incubated for 60 min at 25 °C, protected from light. The sample background control was prepared using the same method as described above. A total of 20 μL of stop solution was added to all background controls before the 60 min incubation period, except for all aspirates, which were added upon completion of the incubation period. The absorbance of the aspirates and background controls was measured at a wavelength of 405 nm using a microplate reader (BioTek Epoch, Winooski, USA) installed with Gen5™ Data Analysis Software. The optical density values for each calibrator against the corresponding concentration of ALP were plotted onto an excel spreadsheet to produce a standard curve. The ALP concentration in aspirates was extrapolated using a linear equation calculated from the standard curve.

### 2.8. Mineralization Staining Assay

Alizarin-red (AR) (Cat:130-22-3, Sigma-Aldrich, Darmstadt, Germany) staining was completed on cells seeded on all composite scaffolds at days 7 and 14 using our established method [25]. A culture plate containing scaffolds with cells were incubated at approximately 15–20 min with 1 mL of AR solution at room temperature. Subsequently, the plate was continuously washed with PBS (1X) to remove any excess stain. Cetylpyridinium chloride 1 mL (10%, Sigma AB, Malmö, Sweden) was added to elute the stain from each scaffold, after 25 min of incubation. The optical density was read (560 nm) using a spectrophotometer (BioTek, Winooski, VT, USA) installed with Gen5™ Data Analysis Software. A standard curve was plotted to measure the intensity of dye stained for each group.

### 2.9. Gene Expression

The total RNA from seeded cells was isolated at days 21 to monitor osteogenic gene expression using our established gene expression protocol [22]. The materials were washed with phosphate buffer solution and then treated with versene solution and incubated for 2–3 min (Invitrogen, Waltham, MA, USA), followed by 15 s vortexing. Then, RNA was isolated using a RNeasy Mini Kit, according to the instructions in manufacturer’s protocol (Cat: 74004, Qiagen, Germantown, MD, USA). Then, cDNA was obtained using 1 μg of a RNA Superscript III First-Strand Synthesis Kit (Cat: 18080051, Invitrogen, Waltham, MA, USA). The q-PCR primers for genes including housekeeping (beta-catenin), bone morphogenic protein 2 (BMP-2), alkaline phosphatase (ALP), type-1 collagen (Col1), runt-related transcription factor 2 (RUNX2), osteopontin (OPN), bone gamma-carboxy glutamic acid-containing protein (BGLAP) were designed using National Centre for Biotechnology Information (NCBI database USA) (Table 1). The cDNA products were amplified with 40 PCR cycles after an initial denaturation step at 95 °C for 3 min, consisting of a denaturation step at 95 °C for 30 s, an annealing temperature ranging from 50 to 60 °C, and an extension step at 72 °C for 5 min. A CFX96 2.0 manager (Bio-Rad,Hercules, CA, USA) was used to calculate relative quantification. The level of expression of each gene stabilized to the housekeeping gene was demonstrated as relative fold expression.

### 2.10. Immunocytochemistry (ICC)

Fabricated scaffolds seeded cells on day 7 were fixed with 4% (*w*/*v*) paraformaldehyde (PFA) (Sigma, Ronkonkoma, New York, NY, USA) in PBS (1X) at room temperature (15 min) and later stained using a published protocol [22]. For primary antibodies, collagen I (Col1) (mouse anti-human collagen 1 monoclonal antibody) (1/2000; Cat: ab6308, Waltham, MA, USA) and rabbit anti-human BMP-2 specific monoclonal antibody for bone morphogenetic protein-2 (BMP2) (1/1000; Cat: ab6285, Abcam, Waltham, Massachusetts, USA) were used. The goat polyclonal secondary antibody to mouse IgG conjugated with Alexa Fluor^®^ 488 (1:500; Cat: ab150113, Abcam, Waltham, MA, USA) was used and counterstained with Hoechst 33342 (ReadyProbes™ Reagent, Waltham, MA, USA) staining. The confocal signals were observed under a confocal microscope (Leica TCS SP5 II, Linford Wood, UK), and images were captured to measure the florescence intensity using our established method [22]. Data were presented as mean ± standard deviation (SD).

### 2.11. Statistics

Data generated from the in vitro experiments were shown as mean ± Standard deviation. The normal distribution of data was confirmed using the Shapiro-Wilk normality test (*p* > 0.05). One-way ANOVA followed by Post-Hoc LSD test was performed using SPSS (v.24, IBM, New York, NY, USA) (Significance level set at *p* < 0.05). 

## 3. Results

### 3.1. Cell Attachment Analysis

An improved cell attachment is an indication of the superior biocompatibility property of a biomaterial scaffold [26]. The attachment of hBMSCs seeded in GC-0 CS, GC-10 CS, GC-20 CS, and GC-40 CS scaffolds was evaluated using SEM imaging. The SEM micrographs captured at 50× magnification demonstrate the porous structure of all composite scaffolds (Figure 1a,c,e,g). Cell attachment was evidenced in all composite scaffolds (Figure 1b,d,f,h), and the cells demonstrated fibroblast-like appearance, as indicated by the red arrow. However, higher attachment was observed on GC-40 CS scaffold, signifying that it had a higher surface hydrophilicity compared to other composite materials. 

### 3.2. Chemical Structure

The broad bands (Figure 2) at peak 3320 cm ^−1^ are related to the -N.H., and the -O.H. is the stretching vibration of the polysaccharide molecules. The absorbance peaks related to the C-H stretching vibration and C-O stretching vibration are around 2920 cm ^−1^ and 1022 cm ^−1^, respectively. These materials, showing peaks displaying −C=O symmetric and asymmetric stretching vibrations, are related to carboxylate (COO^−^) ions at ~1410 and ~1600 cm^−1^. The shifting of peaks at lower wavelengths was noted in calcium silicate materials and could be related to the electrostatic reaction of carboxylate functional groups and calcium silicate. The FTIR spectra bands at 900 and 930 cm^−1^ of calcium-silicate-reinforced gellan-chitosan materials were not in the gellan-chitosan ranges and were related to Si-O and OH stretching and symmetric (Si-O) band stretching, respectively [2,17]. The peak of ~560 cm^−1^ corresponds to the Si-O-Si bonds.

### 3.3. Phase Identification

The XRD spectra of GC-0 CS, GC-10 CS, GC-20 CS, and GC-40 CS scaffolds showed reflections (crystalline) in the range of 5–100 (2θ°), with a step width of 0.02° and intensive peaks at 48.63, 47.43, 39.3, 29.26, 36, and 43.2. This can be described (Figure 3) by the fact that intramolecular electrostatic interaction and the intermolecular solid between the gellan gum carboxyl groups and chitosan amino groups support a certain structural regularity in chitosan to form crystalline regions. The XRD spectra of 40% (*w*/*w*) CS in gellan-chitosan composite found after hydrothermal processing at 200 °C for 24 h were reflection peaks (JCPDS card no. 23-0125) and the Xonotlite phase [27].

### 3.4. Specific Surface Area Analysis

The textural properties of GC-0 CS, GC-10 CS, GC-20 CS, and GC-40 CS materials were subjected to N2-sorption isotherms (Table 2). The surface area of materials observed was in the range of 4–17 m^2^/g. The highest BET surface area (16.72 m^2^/g) was observed in the GC-0 CS scaffold, while the GC-10 CS, GC-20 CS, and GC-40 CS scaffolds demonstrated a surface area in the range of 4.8–6.72 m^2^/g. This information clearly indicates that, except for GC-0 CS material, the particles of the materials were markedly aggregated together. It can be seen (Figure 4a–d,) that the adsorption-desorption (N2) of isotherms exhibits characteristic plots that correspond to type IV isotherms. The mean pore dimensions of GC-10 CS and GC-20 CS materials were found to be in the range of 5–7 nm, whereas the GC-0 CS and GC-40 CS materials showed a significant deviation.

### 3.5. Cell Proliferation and Viability

The proliferation and viability of hBMSCs seeded in all groups demonstrated an increasing trend in terms of reduction of Alamar Blue dye (Expressed as a percentage, Figure 5). Compared with day 0, the other time points demonstrated at least a 1.4-fold significant increase in proliferation of the cells (*p* < 0.05) seeded in all scaffolds. When day 14 was compared with day 7, a statistical difference was noted in GC-10 CS (*p* < 0.05), while there was no significant difference noted between days 7, 14, and 21 in all scaffolds. 

### 3.6. Bone Early Marker (ALP)

An ALP test was performed to confirm the pattern of this early marker in cells seeded on GC-0 CS, GC-10 CS, GC-20 CS, and GC-40 CS scaffolds at different points (Figure 6). The ALP release was steadily increased from day 0 to day 14 in GC-0 CS and GC-10 CS scaffolds, indicated by a threefold significant increase over day 0 (*p* < 0.05). In GC-20 CS and GC-40 CS, this increase was 4- to 5-fold significantly higher (*p* < 0.05) when compared to day 0. The days 7 and 14 comparison showed considerable difference in the ALP activity in GC-0 CS, GC-20 CS, and GC-40 CS.

### 3.7. Detection of Calcium Deposits (Alizarin-Red Staining)

The hBMSCs seeded on GC-40 CS scaffold at days 7 and 21, showed a significantly higher calcification/mineralization (*p* < 0.05) (Figure 7). This result indicates that the GC-40 CS scaffold was significant in terms of enhancing calcium deposition.

### 3.8. Osteogenic Gene Expression

The expression of genes including Runx2, osteopontin (OPN), BGLAP/osteocalcin, osteonectin, ALP, BMP2, and Col1, which are associated with pre-osteoblast formation, was observed in hBMSCs seeded in GC-0 CS, GC-10 CS, GC-20 CS, and GC-40 CS scaffolds at day 21 (Figure 8). The cells seeded in the GC-10 CS scaffold showed a response in the RUNX2 expression as compared with cells seeded in the GC-0 CS scaffold. Meanwhile, cells seeded in the GC-20 CS scaffold showed an increase in ALP expression as compared to cells seeded in the GC-0 CS scaffold (*p* < 0.05). Cells seeded in the GC-40 CS scaffold demonstrated an average 1.5-fold significant increase in RUNX2, ALP, BGLAP, BMP, and osteonectin expression as compared with cells seeded in the GC-0 CS scaffold (*p* < 0.05).

### 3.9. Osteogenic Protein Expression

The confocal micrographs and the fluorescence intensity outcome indicated an increasing pattern of Col1 secretion by hBMSCs seeded in GC-0 CS, GC-10 CS, GC-20 CS, and GC-40 CS scaffolds (Figure 9a). It was found that the hBMSCs seeded in GC-20 CS and GC-40 CS scaffolds secreted 16- and 12-fold significantly higher amounts of Col1 as compared with cells seeded in the GC-0 CS scaffold, respectively, (*p* < 0.01) (Figure 9b). This significant secretion of Col1, 10-fold higher than hBMSCs seeded in the GC-20 CS scaffold, was also observed when compared with hBMSCs seeded in the GC-0 CS scaffold (*p* < 0.01) (Figure 9b). Bone Morphogenic Protein-2 secretion was obvious in the microenvironment of hBMSCs seeded in GC-10 CS, GC-20 CS, and GC-40 CS scaffolds as compared with cells seeded in the GC-0 CS scaffold (Figure 9b). However, a 14-fold significantly higher amount of BMP-2 was observed only at the adjacent cells seeded in the GC-40 CS scaffold as compared with cells seeded in the GC-0 CS scaffold (*p* < 0.01) (Figure 9b)

## 4. Discussion

The selection of cell type is an important factor to be considered when testing different bone grafts for cytotoxicity, biocompatibility, and osteoinduction properties. An established osteogenic cell isolated from the periosteum is an excellent cell type for this purpose [28]. Recently, mesenchymal stromal cells derived from bone marrow have been identified as optimal since they are the precursors of osteoblasts and possess an excellent proliferative capacity. In the present study, the human bone marrow–derived mesenchymal stromal cells were used, as this phenotype has been noted for its isolation technique and characterization following the Mesenchymal and Tissue Stem Cell Committee of the International Society for Cellular Therapy [24].

The SEM cell attachment observations were consistent with a prior report, where the presence of chitosan fibers enhanced gellan hydrogel’s water-holding capacity, confirming its hydrophilic property [20]. Furthermore, the attachment of the cells on the surface was not significantly different; however, the roughness of the scaffold was superior in the scaffold after addition of GC, which indicated that the scaffold provided a physiological environment for cell attachment. This type of rough surface environment can be observed only in material like chitosan [29]. In our earlier study, Tri-calcium phosphate and fucoidan incorporated chitosan induced osteogenic differentiation in bone marrow stromal cells [30]. However, in the present study, for the first time, we have demonstrated that an increase of concentration of CS in GC could enhance the osteogenic potential of marrow stromal cells into osteoblast-like cells in the absence of an osteogenic medium. Therefore, this gellan-chitosan incorporated with CS scaffold is worth being explored in preclinical study to investigate its ectopic bone formation through implanting in the muscle tissue [31]. 

The crystal-type Xonotlite is a hydrated CS type. The analysis of materials using XRD further shows that the main diffraction peaks related to Xonotlite phase GC materials are similar to those of the pure Xonotlite phase. However, the other reflection peaks are related to the GC composite sample. Chitosan employed in bone tissue engineering has received great attention because of its bone regeneration properties and biocompatibility, as demonstrated in pre-clinical animal studies [20]. GC has been described as displaying cytocompatibility and multiplication of mesenchymal stromal cells after nine days of cell culture. In this study, we incorporated CS (three different concentrations) containing GC scaffolds for the differentiation of bone marrow cells into osteoblast-like cells. It has been reported that the assembly of chitosan and gellan can form complexes of polyelectrolytes [32]. The interaction between these polymeric complexes are controlled by intermolecular bonding [11]. In addition, for excellent bone tissue repair, these polymeric complexes are expected to be presented in 3D format [33]. This led us to expanding the properties of the GC materials through the addition of silicates. Calcium silicate (CaSiO_3_) is made of bioactive ${\mathrm{SiO}}_{3}^{2-}$ and Ca^2+^ ions. In accordance with previous data, the Si element at different concentrations dictates biological roles and induces considerable calcium bone mineralization through the increase of osteoclast activity [34]. Our data also confirmed the early increasing trend in the ALP marker in GC-20 CS and GC-40 CS scaffolds; however, the GC-10 CS scaffold also showed an acute increase until day 14, with little difference observed between these three composite scaffolds. In fact, this positive trend of ALP release from cell-seeded scaffolds, especially GC-20 CS and GC-40 CS, is similar to monolayer culture treated with osteogenic media, as demonstrated in our previously published articles [26]. Pre-clinical studies have proposed that Ca^2+^ ions, leached from materials like calcium silicate, may deliver precursor factors for bone formation [23]. In another pre-clinical study, it was demonstrated that defects in rat skulls that were treated with bone-cell-seeded calcium phosphate showed a 2-fold increase in bone formation after 45 days of post-operation assessment [35]. 

ALP is not considered as a strong marker for osteogenic differentiation; hence, the capacity of GC scaffolds to induce osteogenic differentiation was analyzed using alizarin-red staining results. This revealed that alizarin-red staining (color intensity was quantified using the cetylpyridinium chloride method) and the corresponding quantitative results at day 21 were superior in the GC-0 CS scaffold; however, the fold changes in the GC-40 CS scaffold were higher when compared with day 7. This outcome indicates that CS in scaffolds could rapidly accelerate mineral deposition as compared with scaffolds without CS. Though ALP is one of the essential markers measured as an early indicator of osteogenic differentiation, the pattern of bone biomineralization observed from the results of alizarin-red staining in the forsterite scaffold was slightly different due to variable time points between these two experiments [24]. Despite these two parameters, which were well reported in most of the studies associated with bone tissue engineering scaffolds, the osteogenic potential of the cells seeded in scaffolds is ambiguous without a comprehensive osteogenic gene expression quantification of the seeded cells. 

The commitment of bone-marrow-derived mesenchymal stromal cells into osteoblast-like cells is the key indication of bone growth, especially during bone fracture healing. This typical bone lineage commitment is initiated by a number of putative genes and transcription factors, including runt-related transcription factor 2 (RUNX2) [36]. RUNX2 is an important mark in the osteogenic differentiation process, and it also takes part during bone tissue mineralization [37]. Elevated expression of RUNX2 in cells seeded in the GC-40 CS scaffold clearly indicates that this scaffold is suitable for the osteogenic induction and rapid osteogenic differentiation of seeded cells. However, a significant increase in RUNX2 in cells seeded in GC-10 CS as compared with GC-0 CS is yet to be fully understood. This outcome is in accordance a the reported data related to the osteogenic role of silicate in enhancing human marrow stem cells [38].

Alkaline phosphatase (ALP) is an osteogenic marker responsible for extracellular matrix (ECM) mineralization, and ALP gene expression explains osteogenic differentiation patterns [27]. Elevated ALP gene expression was observed in cells seeded in both GC-20 CS and GC-40 CS scaffolds, and this suggests that, though low amounts of CS are enough to induce early differentiation of cells, a high concentration is essential to induce the expression of mid- and late-stage osteogenic differentiation markers [24,39]. BMP-2 is a 396 amino acid polypeptide that can induce mesenchymal stem cell differentiation in bone and cartilage tissues [40]. BMP2 is a dynamic factor in osteogenesis in mediating the condensation process of cells, and it can induce osteoblastic phenotype differentiation. Research has demonstrated that BMP 2 induced cellular chemotaxis and subsequently orchestrated tissue remodeling during critical-sized bone defect healing [41]. In the current study, a higher gene expression of BMP2 in hBMSCs could have triggered ALP activity and promoted the formation of calcium apatite, which was evidenced in gene expression and alizarin-red staining outcomes. In a recent report, it was demonstrated that the presence of silicate activated BMP2 signaling and upregulation of its downstream SMAD genes. These results could support our outcomes, where the increased concentration of CS could have amplified the overall osteogenic effect of GC [42]. This postulation is also in agreement with the increased immunofluorescence expression of BMP2 by hBMSCs seeded in GC-10 CS, GC-20 CS, and GC-40 CS scaffolds as compared with the GC-0 CS scaffold. However, a 14-fold significantly higher amount of BMP2 was only observed in the cells seeded on the GC-40 CS scaffold as compared with cells seeded on the GC-0 CS scaffold, though it was observed at an earlier time point [43]. Likewise, type I collagen is an organic matrix that can be predominately found on the periphery of bone cells such osteoblasts and osteocytes. Though an increase was detected at an early time point (day 7) in protein expression, its relative significant gene expression was only observed at day 21, which could be due to the fact that it is considered an early marker of osteogenesis [44]. 

The organic part of the bone extracellular matrix is type-1 collagen, and it also contains osteopontin (OPN), osteonectin, and BGLAP. These proteins are commonly used to indicate the osteogenic lineage commitment of mesenchymal stromal cells either in early or late stage [45]. An obvious decline in Col1, OPN, and BGLAP could indicate the osteogenic induction of scaffolds starts at late time points. Early time point data showed osteocalcin could improve osteoblast-like cells’ adherence in bio-cement; it enhances the appearance of active osteoblasts and bone healing around hydroxyapatite/collagen composites [46]. A rise in the levels of BGLAP/osteocalcin is an indication of hydroxyapatite formation during osteogenesis in scaffolds. The changes in the 40% calcium silicate–enforced gellan-chitosan group gene expression activity could be related to the fact that the scaffold has Si, with calcium for a positive impact on bone homeostasis, which was comparatively higher than calcium-silicate-added gellan-chitosan (0%, 10%, 20%). The physical, chemical, structural, and topographical characteristics of the matrix play a vital role in determining the differentiation of progenitor cells into matured bone cells [47]. 

The microarchitecture of materials plays a dynamic role in the differentiation of pre-osteoblast cells. Furthermore, the microstructure can influence the infiltration of cells, the transportation efficiency of nutrients, as well as the extracellular matrix accumulation through growth factors and vascularization [48]. The effect of material pore microarchitecture on the osteogenic potential of human MSCs is worth to be explored further. The hierarchical anatomy of native bone indicated that the pore architecture plays an important role in osteogenic differentiation. In the present study, the effect of pore architecture (i.e., pore dimension and morphology of pore connectivity) on the osteogenic differentiation of hBMSCs is unclear. In another study, it was reported that micropores can enhance the osteogenic-related functions of cells in vitro and the osteogenic activity of scaffolds in vivo. It was postulated that the increased osteogenic differentiation could be due enhanced protein adsorption sites and the accelerated release of degradation products, which facilitates the interactions between scaffolds and cells [49]. In addition, the discrepancy in some gene and protein expressions could be addressed if the parameters are further explored with more than 28 days of incubation and with positive controls such as cell-seeded commercial scaffolds and monolayers treated with osteogenic media.

## 5. Conclusions

An increased percentage of calcium-silicate incorporated scaffolds demonstrated a significant deposition of minerals. Further mineralization reinforced the up-regulation of some osteogenic genes, including BMP2, Runx2, BGLAP/osteocalcin, and osteonectin, and selected protein expression indicated that these GC-CS scaffolds are worth to be explored further in in vivo experiments prior to be considered for translation into human applications, especially to repair critical bone defects.

## Figures and Tables

**Figure 1 polymers-13-03211-f001:**
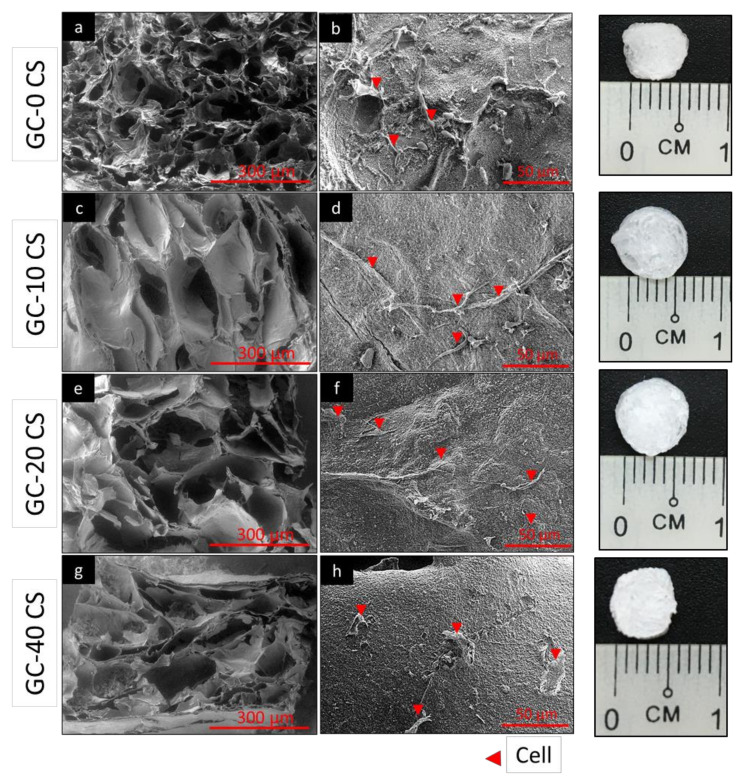
Scanning electron microscopy: (**a**,**b**) GC-0 CS, (**c**,**d**) GC-10 CS, (**e**,**f**) GC-20 CS, and (**g**,**h**) GC-40 CS scaffolds.

**Figure 2 polymers-13-03211-f002:**
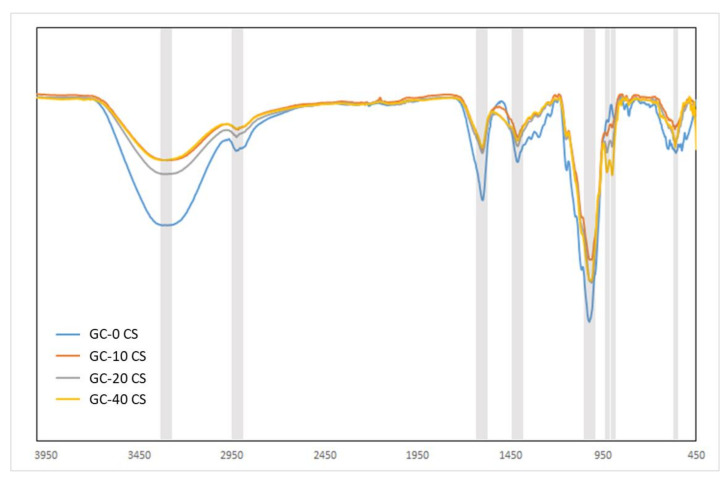
Fourier transmission infra-red spectra of GC-0 CS, GC-10 CS, GC-20 CS, and GC-40 CS scaffolds.

**Figure 3 polymers-13-03211-f003:**
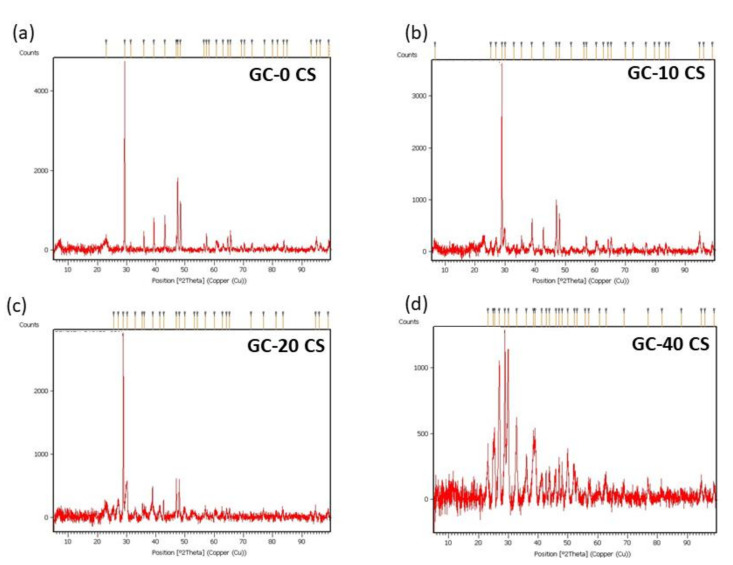
XRD spectra: (**a**) GC-0 CS, (**b**) GC-10 CS, (**c**) GC-20 CS, and (**d**) GC-40 CS scaffolds. The two axes are the position (2θ°-copper) and the counts indicated in the data.

**Figure 4 polymers-13-03211-f004:**
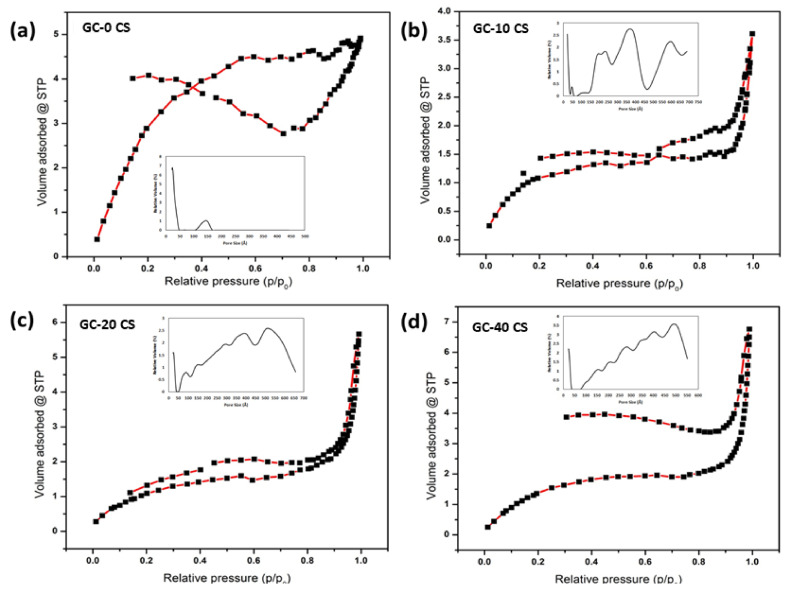
BET analysis: (**a**) GC-0 CS, (**b**) GC-10 CS, (**c**) GC-20 CS, and (**d**) GC-40 CS scaffolds. The two axes are the relative pressure and the volume adsorbed at STP.

**Figure 5 polymers-13-03211-f005:**
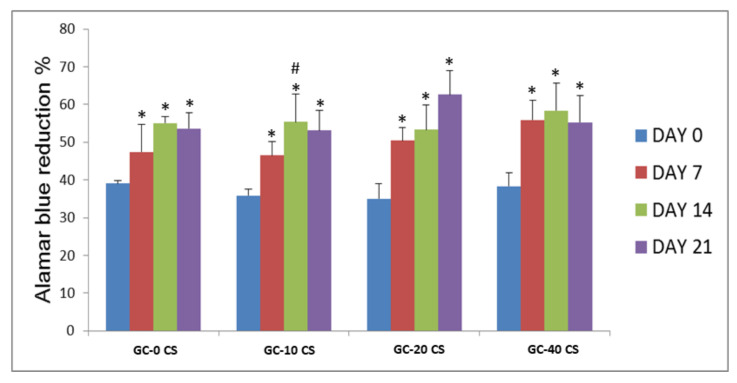
Proliferation and viability (% of Alamar Blue reduction) of hBMSCs seeded in GC-0 CS, GC-10 CS, GC-20 CS, and GC-40 CS scaffolds at days 0, 7, 14, and 21. Statistical analyses: One-way ANOVA followed by Post-Hoc LSD test; * *p* < 0.05 indicates the level of statistical significance between days 7, 14, and 21 against day 0; # *p* < 0.05 indicates comparison between days 7 and 14.

**Figure 6 polymers-13-03211-f006:**
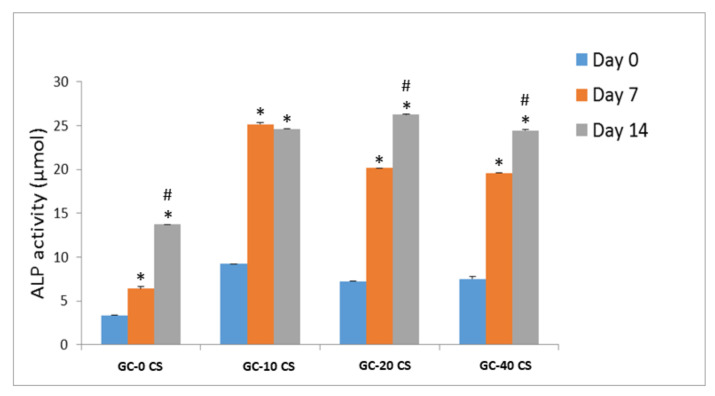
Alkaline phosphatase release of hBMSCs seeded in GC-0 CS, GC-10 CS, GC-20 CS, and GC-40 CS scaffolds at days 0, 7, and 14. Statistical analyses: One-way ANOVA followed by Post-Hoc LSD test; * *p* < 0.05 indicates the level of statistical significance between days 7, and 14 against day 0; # *p* < 0.05 indicates comparison between days 7 and 14.

**Figure 7 polymers-13-03211-f007:**
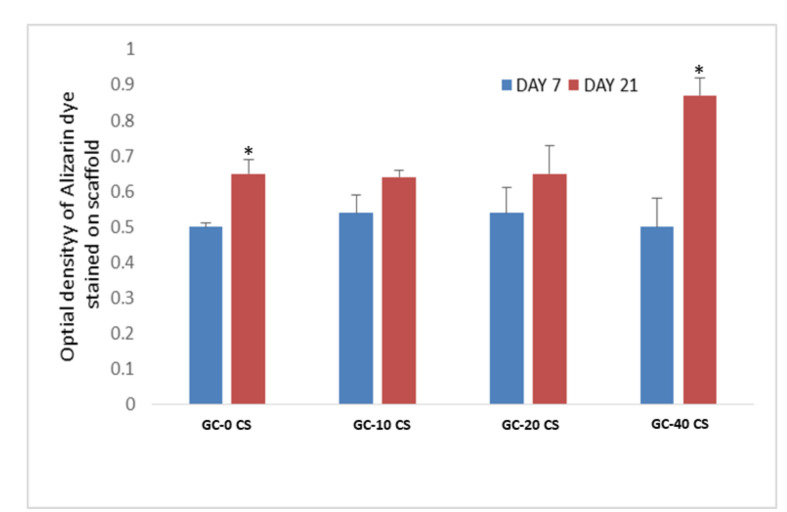
Alizarin-Red staining of hBMSCs seeded in GC-0 CS, GC-10 CS, GC-20 CS, and GC-40 CS scaffolds at days 7 and 21. Statistical analyses: One-way ANOVA followed by Post-Hoc LSD test; * *p* < 0.05 indicates level of statistical significance.

**Figure 8 polymers-13-03211-f008:**
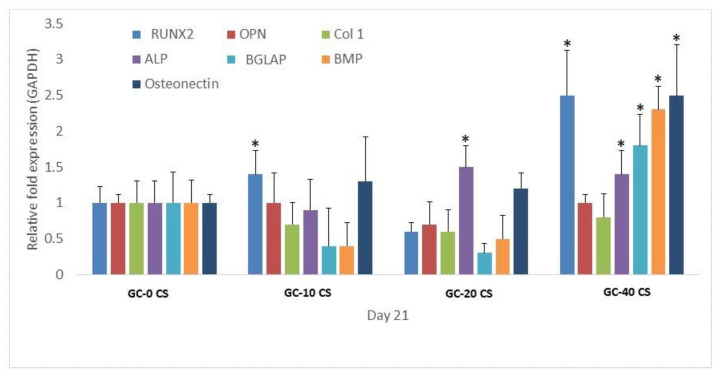
Osteogenic gene expression (Runx2, osteopontin (OPN), type-1 collagen (Col1), alkaline phosphatase (ALP), bone gamma-carboxy glutamic-acid-containing protein (BGLAP), bone morphogenic protein (BMP), osteonectin) of hBMSCs seeded in GC-0 CS, GC-10 CS, GC-20 CS, and GC-40 CS scaffolds at day 21. Statistical analyses: One-way ANOVA followed by Post-Hoc LSD test; * *p* < 0.05 indicates level of statistical significance.

**Figure 9 polymers-13-03211-f009:**
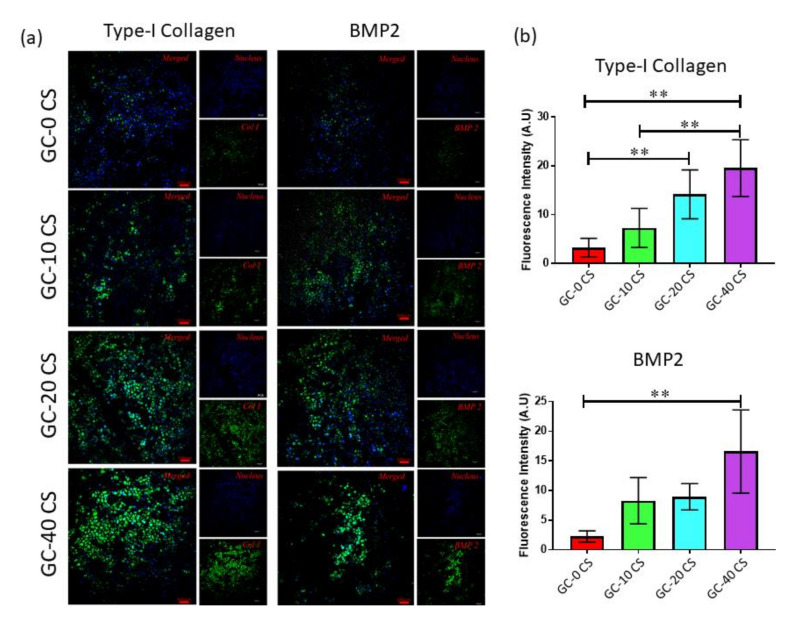
Osteogenic protein expression of hBMSCs seeded in GC-0 CS, GC-10 CS, GC-20 CS, and GC-40 CS scaffolds at day 7: (**a**) Type-I collagen (Col1) and BMP2 confocal images and (**b**) ImageJ semiquantitative analysis of confocal images. Statistical analyses: One-way ANOVA followed by Post-Hoc LSD test; ** *p* < 0.01 indicates level of statistical significance between all groups against GC-0 CS.

**Table 1 polymers-13-03211-t001:** Forward and Reverse Primer Sequences.

Name	Sequence
Col1 Forward	CCCGCAGGCTCCTCCCAG
Col1Reverse	AAGCCCGGATCTGCCCTATTTAT
OPN Forward	CAGCCAGGACTCCATTGACTCGA
OPN Reverse	CCACACTATCACCTCGGCCATCA
BGLAP Forward	GGAGGGCAGCGAGGTAGTGAAGA
BGLAP Reverse	GCCTCCTGAAAGCCGATGTGGT
RUNX2 Forward	CCGCCATGCACCACCACCT
RUNX2 Reverse	CTGGGCCACTGCTGAGGAATTT
BMP2 Forward	TGGCCCACTTGGAGGAGAAACA
BMP2 Reverse	CGCTGTTTGTGTTTGGCTTGACG
ALP Forward	GATGTGGAGTATGAGAGTGACG
ALP Reverse	GGTCAAGGGTCAGGAGTTC
Osteonectin Forward	TTGCAATGGGCCACATACCT
Osteonectin Reverse	GGGCCAATCTCTCCTACTGC

**Table 2 polymers-13-03211-t002:** Analysis of BET for calcium-silicate reinforced Gellan-chitosan scaffold.

S. No.	Sample Name	BET Surface Area m^2^/g	Pore Volume cc/g	Pore Size nm
1	GC-0 CS	16.72	0.0040	19.03
2	GC-10 CS	4.80	0.0052	5.79
3	GC-20 CS	4.63	0.0093	6.99
4	GC-40 CS	6.72	0.0057	41.19

## Data Availability

The authors confirm that the data supporting the findings of this study are available within the article.

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
