# Peer review of "Calcium-Silicate-Incorporated Gellan-Chitosan Induced Osteogenic Differentiation in Mesenchymal Stromal Cells"

_polymers, 2021, doi:10.3390/polym13193211_

Round 1

Reviewer 1 Report

Paper titled (Calcium Silicate Incorporated Gellan-Chitosan Induced Osteogenic Differentiation of Mesenchymal Stromal Cells) by Genasan et al. discussed the formulation of gelatin-chitosan containing calcium and tested its ability to produce osteogenic effect in mesenchymal stromal cells in vitro. This is an interesting study and the data support the conclusion. I have the following recommendations:

1- Introduction need to be spletted out into paragraphs each one delivers certain idea, not to be in a large single paragraph like that in the page 2 &3.

2- Sources of all chemicals should be written completely and consistently (company, town, state, country) + cat number if available

3- The same for softwares + version

4- The same for apparatuses (+ model)

5- Chitosan MWt need to be added.

6- How colorimetric assays were detected? 

7- Many parts in methods come without references

8- part 2.9. mention the type of kits

9- Section 2.11. Did the authors check the normality  of distribution of the data by an appropriate test such as K-S test before applying one-way ANOVA?

10- Fig 5,6,7,8 : please mention the type of the presented data & the applied statistical test in each figure legend.

11 -Fig 6: was there a difference between day 14 & day 7 at any of the preparatons? please indicate you have performed every possible comparison between the groups in methods & revise this figure statistics.

Author Response

Paper titled (Calcium Silicate Incorporated Gellan-Chitosan Induced Osteogenic Differentiation of Mesenchymal Stromal Cells) by Genasan et al. discussed the formulation of gelatin-chitosan containing calcium and tested its ability to produce osteogenic effect in mesenchymal stromal cells in vitro. This is an interesting study and the data support the conclusion. I have the following recommendations:

Many thanks to the reviewer for your effort to addressing the limitations and giving constructive comments/suggestions for the improvement of the quality of the paper for the publication. 

1- Introduction need to be spletted out into paragraphs each one delivers certain idea, not to be in a large single paragraph like that in the page 2 &3.

Thanks for the reviewer’s suggestion. The introduction has been separated into different subparagraphs initiated with its own topic sentence.

2- Sources of all chemicals should be written completely and consistently (company, town, state, country) + cat number if available

Thanks for the reviewer’s suggestions. We have included the source of chemicals as appropriate.

3- The same for softwares + version

Thanks for the reviewer’s suggestions. We have included the source of software and its version as appropriate.

4- The same for apparatuses (+ model)

Thanks for the reviewer’s suggestions. We have included the source of apparatuses  and its model as appropriate.

5- Chitosan MWt need to be added.

Thanks for the reviewer’s suggestions. The MW of the chitosan has been added.

6- How colorimetric assays were detected?

Thanks for the reviewer’s request. The details of the procedure have been added in the section 2.7. Alkaline Phosphatase (ALP) Activity  .

7- Many parts in methods come without references

Thanks for the reviewer’s suggestion. All the methods have been included with index citation of our established and published protocols.

8- part 2.9. mention the type of kits

Thanks for the reviewer’s request. Then RNA isolation was performed using RNeasy Mini Kit, Qiagen Cat: 74004, Qiagen, Germantown, USA). It was included in the part 2.9.

9- Section 2.11. Did the authors check the normality of distribution of the data by an appropriate test such as K-S test before applying one-way ANOVA?

Many thanks to the reviewer for this query. Yes, we had performed Shapiro-Wilk normality test, prior to opt for one-way ANOVA. The p>0.05 where the null hypothesis was accepted (data normally distributed).

10- Fig 5,6,7,8 : please mention the type of the presented data & the applied statistical test in each figure legend.

Many thanks for the reviewer’s suggestion. The type of the presented data and applied statistical tests were stated in the legend of figure 5-9.

11 -Fig 6: was there a difference between day 14 & day 7 at any of the preparations? please indicate you have performed every possible comparison between the groups in methods & revise this figure statistics.

The differences in the fold changes were examined and significance has been included in the graph.

Reviewer 2 Report

The article traces a interesting experimental study about calcium silicate incorporated gellan-chitosan induced osteogenic differentiation of mesenchymal stromal cells. The methods are properly conducted. The availability of data adheres to the expected standards of your research community. The claims are appropriately discussed in the context of previous literature. The manuscript is clearly written. There are no special ethical concerns. Plagiarism detector program showed exceptionally low content similarity (<1.0%), guaranteeing the originality of this article. Article with merit for publication and compatible proposal within the scope of the journal Polymers. Point-to-point instructions for improving the text follow in the comments to the author. After major corrections to the form and content of this version, the manuscript will be ready for publication.

Title:

- The title of the manuscript is succinct, clear and represent the study content.

Abstract:

- The abstract does not present a brief introduction on the subject, knowledge gap in the literature or relevance of the study. In addition, in vitro osteogenesis follow-up times are not specified for a summarized understanding of the results of gene expression, cytochemistry and immunocytochemistry. The new writing must be clear and engaging for the reader.

Section: Introduction

- To reach the journal's worldwide audience and translational relevance, it is interesting discuss just a little about the world panorama about biomaterials for guided bone regeneration (GBR). It is suggested to read article Araújo LK, Antunes GS, Melo MM, Castro-Silva II. Brazilian dentists' perceptions of using bone grafts: an inland survey. Acta Odontol Latinoam. 2020 Dec 1;33(3):165-173. English. PMID: 33523080 (https://pubmed.ncbi.nlm.nih.gov/33523080/). This updated reference discuss useful indicators on the clinical use of alloplastics, composites and other biomaterials classifications around the world, the evolution of the biotechnological market in the area and acceptance by professionals and patients, considering cost-effectiveness of membranes and grafts for GBR. Authors can rely on this information to make the Introduction section more robust and to base the translational theme on the global stage.

Sections: Material and Methods / Results:

- No references were cited in the methodology or results as to the use of positive (such as phenol) or negative (such as polystyrene) controls for the in vitro cytotoxicity test as recommended by ISO 10993-5 standard. Authors should explain better methodology and results, according to the identified issues.

- The authors cultured human bone marrow aspirates cells from 50-70 years old patients and claim to have worked with the population of mesenchymal stromal cells, despite not citing or demonstrating previous flow cytometry or multipotency results that characterize them as such. Reference 20, cited as the protocol used to obtain the cells, only infers that the cells were plastic adherent in DMEM medium supplemented with foetal calf serum, able to form extracellular organic bone matrix and with mineralization capacity. Only these criteria are biologically insufficient to state that it is a mesenchymal population, at a very low rate in elderly patients, at most being able to refer osteoprogenitor cells. Authors should review the concept of cell type to avoid bias in the methodology and interpretation of in vitro osteodifferentiation.

 - Furthermore, the authors did not use control of these cells in a classic osteogenic medium, composed of dexamethazone, ascorbic acid and beta-glycerophosphate. It was not clear why it was not used as a parameter, since the objective is precisely to observe the osteodifferentiation process in vitro, reacting to the tested biomaterials.

Section: Discussion

  • For a better discussion of results from biological characterization of biomaterials, considering times, behaviours and quantities of biomarkers found in tests of cytotoxicity, adhesion, spreading, biocompatibility, biodegradability and osteoconductivity, it is recommended to read the following papers: Lomelino RO, Castro-Silva II, Linhares AB, Alves GG, Santos SR, Gameiro VS, Rossi AM, Granjeiro JM. The association of human primary bone cells with biphasic calcium phosphate (βTCP/HA 70:30) granules increases bone repair. J Mater Sci Mater Med. 2012 Mar;23(3):781-8. doi: 10.1007/s10856-011-4530-1. Epub 2011 Dec 27. PMID: 22201029. (https://pubmed.ncbi.nlm.nih.gov/22201029/); Castro-Silva II, Zambuzzi WF, de Oliveira Castro L, Granjeiro JM. Periosteal-derived cells for bone bioengineering: a promising candidate. Clin Oral Implants Res. 2012 Oct;23(10):1238-42. doi: 10.1111/j.1600-0501.2011.02287.x. Epub 2012 Jan 4. PMID: 22221259. (https://pubmed.ncbi.nlm.nih.gov/22221259/)
  • Interestingly, if the tested material can induce in vitro osteodifferentiation of undifferentiated cells without an osteoinductive medium, the authors should further explore the evidence in the literature on osteoinduction of biomaterials confirmed by ectopic tests, in subcutaneous or muscle tissue. 

Section: Conclusions

- Appropriate. The authors summarize the research findings and describe the inherent limitation of translational biomaterials research, which requires prior animal testing, prior to application in humans.

References:

- Many references are out of the journal's standard. Review this section completely.

- The article presents moderately updated references. Among the 50 references, 28 or 56% of this total was published in the last 5 years (2016-2021). If possible, use the most current articles to ensure the publication's originality.

Author Response

The article traces a interesting experimental study about calcium silicate incorporated gellan-chitosan induced osteogenic differentiation of mesenchymal stromal cells. The methods are properly conducted. The availability of data adheres to the expected standards of your research community. The claims are appropriately discussed in the context of previous literature. The manuscript is clearly written. There are no special ethical concerns. Plagiarism detector program showed exceptionally low content similarity (<1.0%), guaranteeing the originality of this article. Article with merit for publication and compatible proposal within the scope of the journal Polymers. Point-to-point instructions for improving the text follow in the comments to the author. After major corrections to the form and content of this version, the manuscript will be ready for publication.

Many thanks to the reviewer for the compliments and your effort to addressing the limitations and giving constructive comments/suggestions for the improvement of the quality of the paper for the publication. 

Title:

- The title of the manuscript is succinct, clear and represent the study content.

Many thanks to the reviewer for the compliments.

Abstract:

- The abstract does not present a brief introduction on the subject, knowledge gap in the literature or relevance of the study. In addition, in vitro osteogenesis follow-up times are not specified for a summarized understanding of the results of gene expression, cytochemistry and immunocytochemistry. The new writing must be clear and engaging for the reader.

Many thanks to the reviewer for the suggestions. The intro statement has been included in the abstract highlighting the problem statement. The time point for the osteogenic study has been included for the gene expression, Alizarin red assay and immunocytochemistry.

Section: Introduction

- To reach the journal's worldwide audience and translational relevance, it is interesting discuss just a little about the world panorama about biomaterials for guided bone regeneration (GBR). It is suggested to read article Araújo LK, Antunes GS, Melo MM, Castro-Silva II. Brazilian dentists' perceptions of using bone grafts: an inland survey. Acta Odontol Latinoam. 2020 Dec 1;33(3):165-173. English. PMID: 33523080 (https://pubmed.ncbi.nlm.nih.gov/33523080/). This updated reference discuss useful indicators on the clinical use of alloplastics, composites and other biomaterials classifications around the world, the evolution of the biotechnological market in the area and acceptance by professionals and patients, considering cost-effectiveness of membranes and grafts for GBR. Authors can rely on this information to make the Introduction section more robust and to base the translational theme on the global stage.

Many thanks to the reviewer for the great recommendation. The useful information from the suggested article has been incorporated in our introduction and the article has been cited accordingly. 

Sections: Material and Methods / Results:

- No references were cited in the methodology or results as to the use of positive (such as phenol) or negative (such as polystyrene) controls for the in vitro cytotoxicity test as recommended by ISO 10993-5 standard. Authors should explain better methodology and results, according to the identified issues.

Many thanks to reviewer for the suggestion. I totally agree to reviewer point for the consideration of positive and negative controls in the study. 

In the current state, the comparison was made to choose the best composition of the calcium silicate in gellan-chitosan. Therefore, the comparison was made between the groups containing different % of calcium silicate.

We will consider the positive and negative controls follow the ISO  10993-5 standard for our next stage of study involving in preclinical animal study.

The authors cultured human bone marrow aspirates cells from 50-70 years old patients and claim to have worked with the population of mesenchymal stromal cells, despite not citing or demonstrating previous flow cytometry or multipotency results that characterize them as such. Reference 20, cited as the protocol used to obtain the cells, only infers that the cells were plastic adherent in DMEM medium supplemented with foetal calf serum, able to form extracellular organic bone matrix and with mineralization capacity. Only these criteria are biologically insufficient to state that it is a mesenchymal population, at a very low rate in elderly patients, at most being able to refer osteoprogenitor cells. Authors should review the concept of cell type to avoid bias in the methodology and interpretation of in vitro osteodifferentiation.

Many thanks to reviewer for highlighting this concept.  The hBMSCs isolation and its characterization on passage-1 to confirm the plastic adherent, phenotypic surface marker expression (lineage positive: CD44, CD73, CD90 and CD105 and lineage negative: CD45 and CD34) and try-lineage differentiation potential (Osteogenic, adipogenic and chondrogenic) were established in our laboratory.

Reference:

Krishnamurithy G,  et al., The physicochemical and biomechanical profile of forsterite and its osteogenic potential of mesenchymal stromal cells. PLoS One. 2019 Mar 27;14(3):e0214212. doi: 10.1371/journal.pone.0214212. PMID: 30917166; PMCID: PMC6436741.

Furthermore, the authors did not use control of these cells in a classic osteogenic medium, composed of dexamethazone, ascorbic acid and beta-glycerophosphate. It was not clear why it was not used as a parameter, since the objective is precisely to observe the osteodifferentiation process in vitro, reacting to the tested biomaterials.

Thanks for the excellent query from the reviewer. Basically we referenced our published articles where we used monolayer treated with osteogenic media and these articles are discussed in the discussion and well cited.

Reference

1.      Krishnamurithy, G., et al. "Characterization of bovine-derived porous hydroxyapatite scaffold and its potential to support osteogenic differentiation of human bone marrow derived mesenchymal stem cells." Ceramics International 40.1 (2014): 771-777.

2.      Krishnamurithy, Genasan, et al. "Proliferation and osteogenic differentiation of mesenchymal stromal cells in a novel porous hydroxyapatite scaffold." Regenerative medicine 10.5 (2015): 579-590.

Section: Discussion

For a better discussion of results from biological characterization of biomaterials, considering times, behaviours and quantities of biomarkers found in tests of cytotoxicity, adhesion, spreading, biocompatibility, biodegradability and osteoconductivity, it is recommended to read the following papers: Lomelino RO, Castro-Silva II, Linhares AB, Alves GG, Santos SR, Gameiro VS, Rossi AM, Granjeiro JM. The association of human primary bone cells with biphasic calcium phosphate (βTCP/HA 70:30) granules increases bone repair. J Mater Sci Mater Med. 2012 Mar;23(3):781-8. doi: 10.1007/s10856-011-4530-1. Epub 2011 Dec 27. PMID: 22201029. (https://pubmed.ncbi.nlm.nih.gov/22201029/); Castro-Silva II, Zambuzzi WF, de Oliveira Castro L, Granjeiro JM. Periosteal-derived cells for bone bioengineering: a promising candidate. Clin Oral Implants Res. 2012 Oct;23(10):1238-42. doi: 10.1111/j.1600-0501.2011.02287.x. Epub 2012 Jan 4. PMID: 22221259. (https://pubmed.ncbi.nlm.nih.gov/22221259/)

Interestingly, if the tested material can induce in vitro osteodifferentiation of undifferentiated cells without an osteoinductive medium, the authors should further explore the evidence in the literature on osteoinduction of biomaterials confirmed by ectopic tests, in subcutaneous or muscle tissue.

Many thanks to the reviewer for suggesting these two excellent articles to be included within the content of our manuscript. We have cited both articles as per your suggestion in the discussion. In fact, the need for the ectopic bone formation test is well stated in the discussion.

Section: Conclusions

- Appropriate. The authors summarize the research findings and describe the inherent limitation of translational biomaterials research, which requires prior animal testing, prior to application in humans.

Many thanks to the reviewer for the compliments.

References:

- Many references are out of the journal's standard. Review this section completely.

- The article presents moderately updated references. Among the 50 references, 28 or 56% of this total was published in the last 5 years (2016-2021). If possible, use the most current articles to ensure the publication's originality

Many thanks to the reviewer for the detailed vetting on our references. We have updated the latest (2016-2021) and most relevant articles as per the request and the percentage now is more than 68%. Some of the important articles published before 2016 are still retained for the references.  

Round 2

Reviewer 1 Report

Thanks for the authors for improving the manuscript, I wish they also address these points:

1- Section 2.5: give the brief method

2- My previous comment on Fig 5, 6 and 7was not addressed. Please compare the day 7 & 14 results

Author Response

1- Section 2.5: give the brief method

Brief Method include

2- My previous comment on Fig 5, 6 and 7was not addressed. Please compare the day 7 & 14 results

The figure 5 and 6 comparisons between day 7 and 14 have been highlighted. Figure 7 no significance noted and Figure 8 only one time point 21 day was performed so no comparison was required

Reviewer 2 Report

The authors provided adequate responses to all requirements identified point-by-point in the original text, its references and figures.

Congratulations to the authors for the significant improvement of the content and format of the work. In this latest version, I consider the article suitable for publication.

Author Response

Thank you for your comment. 

Round 3

Reviewer 1 Report

Minor change is needed :

1- Statistical analyses: Statistical analyses: One way396 ANOVA and Post-Hoc LSD on continuous data, : remove the repeated words

Post-Hoc LSD : does not make sense; Should be : ANOVA followed by Post-Hoc LSD test. Correct this in all figure legends

2- a*p<0.05 indicates the comparison with day 7 and day 14.: should be: a*p<0.05 indicates significant difference between day 7 and day 14 results. correct allover the paper please.

3- please improve the figure appearance by increasing the X & Y axis thicknesses & sharpness of writing or size as the current format is not satisfactory